# Robust Aggregation With Adversarial Experts

## Abstract

We consider a robust aggregation problem in the presence of both truthful and adversarial experts. The truthful experts will report their private signals truthfully, while the adversarial experts can report arbitrarily. We assume experts are marginally symmetric in the sense that they share the same common prior and marginal posteriors. The rule maker needs to design an aggregator to predict the true world state from these experts' reports, without knowledge of the underlying information structures or adversarial strategies. We aim to find the optimal aggregator that outputs a forecast minimizing regret under the worst information structure and adversarial strategies. The regret is defined by the difference in expected loss between the aggregator and a benchmark who aggregates optimally given the information structure and reports of truthful experts.

We focus on binary states and reports. Under L1 loss, we show that the truncated mean aggregator is optimal. When there are at most k adversaries, this aggregator discards the k lowest and highest reported values and averages the remaining ones. For L2 loss, the optimal aggregators are piecewise linear functions. All the optimalities hold when the ratio of adversaries is bounded above by a value determined by the experts' priors and posteriors. The regret only depends on the ratio of adversaries, not on their total number. For hard aggregators that output a decision, we prove that a random version of the truncated mean is optimal for both L1 and L2. This aggregator randomly follows a remaining value after discarding the $k$ lowest and highest reported values. We evaluate our aggregators numerically in an ensemble learning task. We also obtain negative results for general adversarial aggregation problems under broader information structures and report spaces.

## CCS Concepts

• **Theory of computation → Algorithmic game theory**; **Algorithmic mechanism design**.

## Keywords

Robust Aggregation, Adversary, Information Aggregation

**ACM Reference Format:**
Anonymous Author(s). 2018. Robust Aggregation With Adversarial Experts. In *Proceedings of Make sure to enter the correct conference title from your rights confirmation emai (Conference acronym 'XX).* ACM, New York, NY, USA, 15 pages. https://doi.org/XXXXXXX.XXXXXXX

## 1 Introduction

You are a rule maker tasked with aggregating the scores of five judges to assign a final score for an athlete's performance. There is a crucial twist: some of those scores might be tainted by bribes! The briber's motive is unknown, potentially inflating or deflating the score. You have no clue about the underlying details. How should you decide the aggregation rules?

Such concerns of aggregating information with adversarial "experts" also exist in various scenarios. For instance, when the jury debates, some jurors may be swayed by a bribe to deliver a specific verdict. When miners are asked to reach a consensus on a blockchain network, some malicious miners manipulate validations for personal gain. Furthermore, in ensemble learning, when combining predictions from multiple models, some models are compromised by adversarial actors. Therefore, it is crucial to design aggregation rules that are robust to adversarial attacks.

Intuitively, the truncated mean, which discards some highest and lowest scores and averages the remaining scores, seems reasonable and is widely used in practice. But the understanding of its theoretical effectiveness is limited. A natural question is, is this the most effective strategy?

To answer this, we need a clear evaluation criterion for aggregation methods. A common approach is average loss, which calculates the average difference between the aggregator's output and the true state across various cases. Two key elements define a case: the information structure and the adversarial behavior. The information structure is the joint distribution of experts' private signals and the true state. However, average loss heavily depends on the specific set of cases chosen and contradicts the assumption that the adversarial behavior can be arbitrary.

Another option is the worst-case loss, focusing on the maximum loss the aggregator obtains under any case. However, if all experts are completely uninformed, no aggregator can perform well. Therefore, the worst-case loss cannot differentiate between aggregators.

Instead, we adopt a robust framework commonly used in online learning and robust information aggregation [2]. This framework aims to minimize the aggregator's "regret" in the worst case. Regret measures the difference between the aggregator's performance and an omniscient aggregator who knows the information structure and truthful reports.

The framework can be understood as a zero-sum game between two players. Nature chooses an information structure and adversarial strategy, aiming to maximize the regret. The rule maker picks an aggregator to minimize the regret. Generally, solving such a minimax problem is challenging due to the vast action space. With a delicate analysis, prior studies proved that among aggregators who output decisions, the random dictator is optimal without adversarial experts under L1 loss, which implies that the optimal aggregator that outputs probability is simple averaging [3]. However, this analysis does not extend to other scenarios, such as L2 loss. We show

that without adversarial experts, the problem under L2 loss lacks a simple solution.

Paradoxically, introducing adversarial experts does not complicate the problem in some scenarios, it even simplifies it! Regarding the soft aggregators that output a probability, we discover that with a bounded proportion of adversaries, the simple truncated mean is optimal under L1 loss. Furthermore, in the adversarial setting with L2 loss, we provide a closed-form solution, which is unattainable in the non-adversarial setting. Both optimal aggregators fall within the family of piecewise linear functions. For hard aggregators that output a decision, a random version of the truncated mean is optimal for both L1 and L2 loss. The key insight is that the presence of at least one adversarial expert guarantees the existence of equilibria with simple formats, which easy to construct. These equilibria enable us to design optimal aggregators with closed-form formulas.

In summary, we introduce a novel setting that considers adversarial experts in robust information aggregation. This framework enables us to theoretically prove the optimality of the commonly used truncated mean method under L1 loss and provide optimal aggregators under L2 loss, which are piecewise linear.

## 1.1 Summary of Results

*Theoretical Results.* In the original non-adversarial setting in [3], each expert will receive and report a binary private signal, either $L$ (low) or $H$ (high), indicating the binary world state $\omega \in \{0, 1\}$. The experts are marginally symmetric and truthful. That is, they share the same marginal distribution of signals and will report their private signals truthfully. However, correlations may exist between the signals, thus the joint distribution is not determined. The information structure $\theta \in \Theta$ is defined by the joint distribution of private signals and world state.

We extend the above setting to the adversarial setting. In addition to truthful experts, adversarial experts exist and report arbitrarily from the signal set $\{H, L\}$. We assume the adversaries can observe the reports of others and collude. The adversarial strategy is denoted by $\sigma \in \Sigma$.

The rule maker aims to find the optimal aggregator $f$ to solve the minimax problem:

$$R(\Theta, \Sigma) = \inf_{f} \sup_{\theta \in \Theta, \sigma \in \Sigma} \mathbb{E}_\theta[\ell(f(\boldsymbol{x}), \omega) - \ell(opt_\theta(\boldsymbol{x}_T), \omega)].$$

We analyze the problem in two contexts: 1) soft: $f$'s output is a forecast in $[0, 1]$; 2) hard: $f$'s output is a decision in $\{0, 1\}$. The aggregator $f$ can be randomized in the sense that its output is random. $opt$ is the benchmark, which is an omniscient aggregator that knows the underlying information structure $\theta$ of truthful experts and minimizes the expected loss. $\boldsymbol{x}$ is the reports of all experts, $\boldsymbol{x}_T$ is the reports of truthful experts, and $\ell$ is a loss function. In this paper, we discuss two types of loss function, the L1 loss $\ell_1(y, \omega) = |y - \omega|$ and the L2 loss $\ell_2(y, \omega) = (y - \omega)^2$. The benchmark function $opt$ should report the maximum likelihood under L1 loss and the Bayesian posterior under L2 loss. Thus L1 loss is more suitable when we want to output decisions and L2 loss is preferred for probabilistic forecasts. Suppose there are $n$ experts in total and $k = \gamma n$ adversarial experts. The theoretical results are

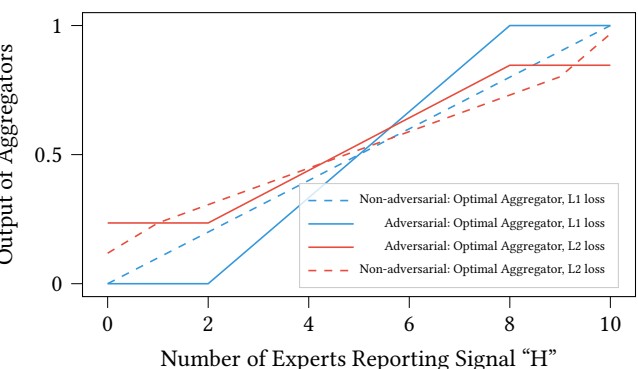

Figure 1: Illustration of optimal soft aggregators for binary aggregation.

shown in Table 1. The optimal soft aggregators are deterministic, and illustrated in Figure 1.

| Loss Function | Experts Model | Regret† | Closed-form? |
|---|---|---|---|
| L1 Loss | Non-adversarial | $c$ | Yes [3] |
| | Adversarial | $c + O(\frac{\gamma}{1-2\gamma})$ | Yes |
| L2 Loss | Non-adversarial | $c + O(\frac{1}{n})$ | No, $O(\frac{1}{\epsilon})^\star$ |
| | Adversarial | $c + O(\frac{\gamma}{(1-2\gamma)^2})$ | Yes |

† $c$ is a constant depending on the prior and the posteriors given signals. $\gamma$ is the ratio of adversarial experts and $n$ is the number of experts.

★ The complexity for computing the $\epsilon$-optimal aggregator $f_\epsilon$, i.e., $\max_{\theta \in \Theta} \mathbb{E}_\theta[\ell(f_\epsilon(\boldsymbol{x}), \omega) - \ell(opt_\theta(\boldsymbol{x}_T), \omega)] \leq \min_f \max_{\theta \in \Theta} \mathbb{E}_\theta[\ell(f(\boldsymbol{x}), \omega) - \ell(opt_\theta(\boldsymbol{x}_T), \omega)] + \epsilon$.

Table 1: Overview of main results.

- **L1 Loss Setting**
  - **Adversarial.** We prove that under the L1 loss, the optimal soft aggregator is $k$-truncated mean, where $k$ is the number of adversarial experts. That is, we drop $k$ lowest reports and $k$ highest reports, then average the remaining reports. Our analysis also reveals that the regret increases asymptotically linear with adversarial ratio $\gamma$ for small $\gamma$. Moreover, the regret is independent of the number of experts $n$.
- **L2 Loss Setting**
  - **Non-adversarial.** In this setting, the optimal soft aggregator remains piecewise linear, with separation points at $\{1, n - 1\}$. Although we do not obtain the closed-form for the optimal aggregator, we prove that we can compute the $\epsilon$-optimal aggregator within $O(1/\epsilon)$

time. Unlike the L1 loss, the regret under the L2 loss increases with $n$. Compared to the L1 loss, the aggregator under the L2 loss is more conservative (i.e., closer to 1/2).

- **Adversarial.** We provide a closed-form expression for the optimal soft aggregator which is a hard sigmoid function with separation points at $\{k, n-k\}$ for small $\gamma$. The regret also increases asymptotically linear with parameter $\gamma$ and is independent of the number of experts $n$. Interestingly, if we set $\gamma = 0$, the formula of the adversarial optimal aggregator cannot match the non-adversarial optimal aggregator. The reason is when $\gamma > 0$, we can construct an equilibrium $(f, \theta, \sigma)$ such that the information structure $\theta$ has a zero loss benchmark. But when $\gamma = 0$, we cannot construct an equilibrium $(f, \theta)$ in the same way. The worst information structure in the equilibrium may have a non-zero loss benchmark, which leads to a more complex optimal aggregator.

We also analyze optimal hard aggregators in Appendix C. For both L1 and L2 loss, the optimal hard aggregators are random, whose expectation is $k$-truncated mean. We call it $k$-ignorance random dictator. It ignores k lowest-scoring and k highest-scoring experts, then randomly follows one of the remaining experts. It echos the results in [3] that the random dictator is optimal for the non-adversarial setting.

*Numerical Results.* In Section 5, we empirically evaluate the above aggregators in an ensemble learning task, which aggregates the outputs of multiple image classifiers. These classifiers are trained by different subsets of the full training set. We utilize the cifar-10 datasets [25]. Our experiments show that the theoretically derived optimal aggregator outperforms traditional methods like majority vote and averaging, particularly under L2 loss. Under L1 loss, the majority vote also performs well. The reason can be that compared to L1 loss, L2 loss penalizes not only being wrong but also how wrong it is.

*Extension to General Model.* In Appendix E, we extend the binary aggregation to the general model. In detail, we consider a broader range of information structures and experts' report space. We show that a small group of adversaries can effectively attack the aggregator. We introduce a metric to estimate the regret. Intuitively, the metric is the maximum impact $k$ experts can make regarding the benchmark aggregator $opt$. This metric allows us to establish a bound on the minimal regret, with the help of a regularization parameter of information structures.

## 2 Related Work

*Robust Information Aggregation.* Arieli et al. [2] first propose a robust paradigm for the information aggregation problem, which aims to minimize the regret under the worst information structures. They mainly study the conditional independent setting. There is a growing number of research on the robust information aggregation problem. Neyman and Roughgarden [29] also use the robust regret paradigm under the projective substitutes condition and shows that averaging is asymptotically optimal. De Oliveira et al. [10]

consider the robust absolute paradigm and prove that we should pay more attention to the best single information source. Pan et al. [31] consider the optimal aggregator with second-order information. Guo et al. [16] provide an algorithmic method to compute the near-optimal aggregator for the conditional independent information structure. We consider a different set of information structures, and more importantly, the existence of adversaries. We also obtain the exact optimal aggregator with closed forms in the adversarial setting.

Our paper is most relevant to Arieli et al. [3], which considers the symmetric agents setting with the same marginal distribution. They prove the random dictator strategy is optimal under some mild conditions. We extend their results to different loss functions and the adversarial setting. Our results show that the optimal hard aggregator follows a random expert after discarding some values, which extends the random dictator strategy.

*Adversarial Information Aggregation.* In the crowdsourcing field, some works aim to detect unreliable workers based on observed labels. They mainly consider two kinds of unreliable workers, the truthful workers but with a high error rate, and the adversarial workers who will arbitrarily assign labels. One possible approach is using the "golden standard" tasks, which means managers know the ground truth [12, 26, 34]. When there are no "golden standard" tasks, Hovy et al. [20], Jagabathula et al. [21], Kleindessner and Awasthi [24], Vuurens et al. [36] detect the unreliable workers via revealed labels under some behavior models such as the Dawid-Skene model. Other works focus on finding the true labels with adversarial workers. Steinhardt et al. [35] consider a rating task with $\alpha n$ reliable workers, and others are adversarial workers. They showed that the managers can use a small amount of tasks, which is not scaled with $n$, to determine almost all the high-quality items. Ma and Olshevsky [27] solve the adversarial crowdsourcing problem by rank-1 matrix completion with corrupted revealed entries. Other works also focus on the data poison attacks in crowdsourcing platforms [7, 8, 13, 15, 28, 37]. Unlike crowdsourcing, we do not assume the existence of many similar tasks and focus on the one-shot information aggregation task.

Han et al. [17] and Schoenebeck et al. [33] propose a peer prediction mechanism for a hybrid crowd containing both self-interested agents and adversarial agents. Han et al. [17] focus on the binary label setting and propose two strictly posterior truthful mechanisms. Schoenebeck et al. [33] prove their mechanism guarantees truth-telling is a $\epsilon$-Bayesian Nash equilibrium for self-interested agents. The focus of our paper is the aggregation step, thus we do not consider the incentives of non-adversarial agents but assume they are truth-telling.

Kim and Fey [23] study the voting problem when there are voters with adversarial preferences. They prove that it is possible that a minority-preferred candidate almost surely wins the election under equilibrium strategies. They want to determine when the majority vote can reveal the true ground truth while we want to find a robust aggregator for any situation.

*Robust Ensemble Learning.* Ensemble learning methods leverage the power of multiple models to improve prediction accuracy, similar to how aggregating predictions from a diverse crowd can produce better forecasts than individual opinions. The earliest work

of ensemble learning can date back to the last century [9]. They aim to combine different classifiers trained from different categories into a composite classifier. [32] provides a new algorithm to convert some weak learners to strong learners. The most widely-used ensemble learning methods include bagging [5], AdaBoost [18], random forest [6], random subspace [19], gradient boosting [14]. Dong et al. [11] provides a comprehensive review of ensemble learning. The main difference between ensemble learning and information aggregation is that ensemble learning is a training process and thus involves multi-round aggregation. Unlike them, our method does not assume any knowledge about the underlying learning models and only needs the final outputs of models.

In the robust learning field, there are many works aimed at data poison attacks to improve the robustness of learning algorithms [4, 22, 38]. In comparison, our adversaries cannot change the training data, but alter the output of learning models.

## 3 Problem Statement

We define $\{0, 1, \cdots, n\}$ as $[n]$ and $\Delta_X$ as the set of all possible distributions over $X$. For a distribution $P$, $supp(P)$ denotes its support set.

Suppose the rule maker wants to determine the true state $\omega$ from a binary choice set $\Omega = \{0, 1\}$. She is uninformed and asks $n$ experts for advice. Each expert will receive a signal $s_i$ from a binary space $S = \{L, H\}$ (low, high), indicating that with low (high) probability the state is 1. Then they will truthfully report $L$ or $H$ according to their signals. The binary signal assumption can be relaxed, as we can always construct a binary signal structure from a non-binary structure with the same joint distribution over binary reports [3].

Experts share the same prior $\mu = \Pr[\omega = 1]$ and posterior given signals: $p_0 = \Pr[\omega = 1|s_i = L]$ and $p_1 = \Pr[\omega = 1|s_i = H]$. We assume $p_0 < \frac{1}{2} < p_1$, otherwise the signals are not informative (e.g., if $p_0 < p_1 < 1/2$, then any signal is a low signal). We also define their inverse probabilities $a = \Pr[s_i = H|\omega = 1]$, $b = \Pr[s_i = H|\omega = 0]$. The joint distribution of true state and signals $\theta \in \Delta_{\Omega \times S^n}$ is drawn from a family $\Theta$. It is also called the information structure. Since there may exist correlations between the experts' signals, the parameters $\mu, a, b$ alone are insufficient to determine an information structure.

In the non-adversarial setting, the experts are truthful when they report their signals. We could relax this assumption to rational experts who are revenue maximizers by applying proper incentive mechanisms. Following Arieli and Babichenko [1], we assume the experts are anonymous, which can be relaxed due to the symmetry of experts. Thus the rule maker only sees the number of experts reporting $H$, denoted by $x \in [n]$. Then she needs to choose a randomized aggregator $f : [n] \to \Delta_{[0,1]}$, which maps reports to a (possibly random) belief $\in [0, 1]$ about being in state 1. We first focus on randomized soft aggregators and will extend the results to randomized hard aggregators $f : [n] \to \Delta_{\{0,1\}}$ in Appendix C.

In the adversarial setting, there are $k = \gamma n$ adversarial experts who can arbitrarily report from $\{L, H\}$. We assume they are omniscient. That is, they know the true state and reports of other truthful experts and can collude. They will follow a randomized strategy $\sigma \in \Sigma : \Omega \times S^{n-k} \to \Delta_{[k]}$ that maps truthful reports to a (random) number of additional $H$. Suppose the set of truthful experts is $T$,

and the set of adversarial experts is $A$. We use $x_T$ and $x_A$ to represent the number of reports $H$ from truthful and adversarial experts, respectively.

*Robust Aggregation Paradigm.* Given the families of information structures $\Theta$ and strategies $\Sigma$, the rule maker aims to minimize the regret compared to the non-adversarial setting in the worst information structure. That is, the rule maker wants to find the optimal function $f^*$ to solve the following minimax problem:

$$R(\Theta, \Sigma) = \inf_f \sup_{\theta \in \theta, \sigma \in \Sigma} \mathbb{E}_{\theta, \sigma}[\ell(f(x), \omega)] - \mathbb{E}_\theta[\ell(opt_\theta(x_T), \omega)].$$

$\ell$ is a loss function regarding the output of the aggregator and the true state $\omega$. $opt_\theta(\cdot)$ is a benchmark function, outputting the optimal result given the joint distribution $\theta$ and truthful experts' reports $x_T$ to minimize the expected loss: $opt_\theta(x_T) = \arg\min_g \mathbb{E}_\theta[\ell(g(x_T), \omega)]$.

We consider two commonly used loss function, L1 loss $\ell_1(y, \omega) = |y - \omega|$ and L2 loss $\ell_2(y, \omega) = (y - \omega)^2$. L2 loss will punish the aggregator less when the prediction is closer to the true state. Thus it encourages a more conservative strategy for the aggregator to improve the accuracy in the worst case. L1 loss will encourage a radical strategy to approximate the most possible state.

For short, we also define

$$R(f, \theta, \sigma) = \mathbb{E}_{\theta, \sigma}[\ell(f(x), \omega)] - \mathbb{E}_\theta[\ell(opt_\theta(x_T), \omega)],$$

is aggregator $f$'s regret on information structure $\theta$ and adversarial strategy $\sigma$. $R(f, \Theta, \Sigma) = \sup_{\theta \in \Theta, \sigma \in \Sigma} R(f, \theta, \sigma)$ is the maximal regret of aggregator $f$ among the family $\Theta, \Sigma$.

## 4 Theoretical Results

In this section, we analyze the optimal aggregators under different settings theoretically. Due to space limitations, all the proofs in this section are deferred to Appendix B.

### 4.1 L1 loss

We start from L1 loss. In the non-adversarial setting, Arieli et al. [3] prove the optimal hard aggregator is the random dictator, i.e., randomly and uniformly following an expert. With a step further, it reveals that simple averaging $f(x) = x/n$ is the optimal soft aggregator.

In the adversarial setting, we prove that the optimal aggregator is the $k$-truncated mean, when the adversary ratio $\gamma = k/n$ is upper-bounded (Theorem 4.2). $k$-truncated mean discards $k$ lowest and $k$ highest reports, then outputs the average among left reports.

DEFINITION 4.1 ($k$-TRUNCATED MEAN). *We call $f$ is $k$-truncated mean if*

$$f(x) = \begin{cases} 1 & x \geq n - k \\ 0 & x \leq k \\ \dfrac{x - k}{n - 2k} & otherwise \end{cases} \tag{1}$$

THEOREM 4.2. *When*

$$\gamma \leq \min\left(\frac{a}{1+a}, \frac{1-b}{2-b}, \frac{-(1-\mu)b + \mu a}{\mu - (1-\mu)b + \mu a}, \frac{(1-\mu)(1-b) - \mu(1-a)}{(1-b)(1-\mu) + 1 - \mu - \mu(1-a)}\right)$$

*, the $k$-truncated mean is optimal under the L1 loss. Recall that $\mu$ is the prior, $a = \Pr[s_i = H|\omega = 1]$ and $b = \Pr[s_i = H|\omega = 0]$. Moreover,*

*the regret is*

$$R(\Theta, \Sigma) = \frac{(1-\gamma)\,(1-(1-\mu)(1-b)-\mu a)}{1-2\gamma}.$$

Intuitively, when signals are highly informative ($a = Pr[s_i = H|\omega = 1] \approx 1$ or $b = Pr[s_i = H|\omega = 0] \approx 0$), adversaries struggle to manipulate the majority of experts' opinions. This resilience to adversarial attacks results in a less strict bound ($a/(1+a), (1-b)/(2-b) \approx 1/2$). On the other hand, if the signals are less informative and distinguishable ($Pr[s_i = H, \omega = 1] \approx Pr[s_i = H, \omega = 0]$ or $Pr[s_i = L, \omega = 1] \approx Pr[s_i = L, \omega = 0]$), adversaries can more easily distort the results, leading to a tighter bound ($a\mu - b(1-\mu) \approx 0$ or $\mu(1-a) - (1-\mu)(1-b) \approx 0$).

When $\gamma$ is sufficiently large, the $k$-truncated mean may not be optimal. Nonetheless, we can infer by the same argument that the optimal aggregator will be a constant, either 1 or 0. It means that the aggregator is uninformative regardless of experts' reports.

*Proof Sketch.* We prove Theorem 4.2 in two steps.

- Lower Bound: We first construct a bad case, including the information structure and adversarial strategy $\theta_b, \sigma_b$, which establishes a lower bound $R$ of the regret for any aggregator.
- Upper Bound: We construct an aggregator—the $k$-truncated mean. On the one hand, it matches the lower bound $R$ under the bad case $\theta_b, \sigma_b$. On the other hand, we prove that it possesses some special properties, and therefore, the worst-case scenario it corresponds to is $\theta_b, \sigma_b$. Thus we prove the optimality of the $k$-truncated mean.

*If the Number of Adversaries $k$ is Unknown.* To identify the optimal aggregator, it is crucial to know the parameter $k$. In practice, the exact number of adversaries may be unable to know. Instead, We may only obtain an estimator $k'$ of $k$. In this case, Lemma 4.3 shows that the regret grows asymptotically linear with the additive error $|k - k'|$.

**LEMMA 4.3.** *Suppose the number of adversaries is $k$, then for any $k'$, the $k'$-truncated mean obtains the regret*

$$R(f_{k'}, \Theta, \Sigma) = \frac{k' - k + (n-k)\,(1-(1-\mu)(1-b)-\mu a)}{n - 2k'}.$$

## 4.2 L2 loss

We first show that without some prior knowledge of the information structure, we cannot obtain non-trivial optimal aggregator. We then show that with some partial prior knowledge, the optimal aggregators are non-trivial.

*4.2.1 Unknown Prior.* In the L1 loss setting, the optimal aggregator is independent with three parameters we defined before, the prior $\mu$ and the marginal report distributions $a, b$. However, in the L2 loss setting, the optimal aggregator also depends on these parameters. When these parameters are unknown to the rule maker, it is impossible to obtain any informative aggregator (Lemma 4.4, Lemma 4.5).

**LEMMA 4.4.** *When $\mu, a, b$ is unknown, the random guess is optimal. Formally, let $(\Theta_{\mu,a,b}, \Sigma_{\mu,a,b})$ includes all information structures and adversarial strategies with parameters $\mu, a, b$ and $(\Theta, \Sigma) =$*

$\bigcup_{0 \le \mu \le 1, a > b}(\Theta_{\mu,a,b}, \Sigma_{\mu,a,b})$. *Then $f^*(x) = 1/2$ and $R(\Theta, \Sigma) = 1/4$. It holds for both adversarial and non-adversarial settings.*

**LEMMA 4.5.** *When $\mu$ is known but $a, b$ is unknown, the prior guess is optimal. Formally, let $(\Theta_{a,b}, \Sigma_{a,b})$ includes all information structures and adversarial strategies with parameters $\mu, a, b$ and $(\Theta, \Sigma) = \bigcup_{a>b}(\Theta_{a,b}, \Sigma_{a,b})$. Then $f^*(x) = \mu$ and $R(\Theta, \Sigma) = \mu(1-\mu)$. It holds for both adversarial and non-adversarial settings.*

We will obtain a non-trivial result when all three parameters are known to the rule maker. In the rest of this section, we assume $\mu, a, b$ are known.

*4.2.2 Partial Prior Knowledge: Non-adversarial Setting.* First, we discuss the non-adversarial setting. Unfortunately, obtaining a closed-form expression for the optimal aggregator is infeasible without solving a high-order polynomial equation. However, we can calculate it efficiently as Theorem 4.6 shows.

**THEOREM 4.6.** *There exists an algorithm that costs $O(1/\epsilon)$ to find the $\epsilon$-optimal aggregator $f_\epsilon$ in the non-adversarial setting. That is, define the maximal regret of $f$, $R(f, \Theta) = \max_{\theta \in \Theta} \mathbb{E}_\theta[\ell(f(x), \omega)] - \mathbb{E}_\theta[\ell(opt_\theta(x), \omega)]$. $f_\epsilon$ is a $\epsilon$-optimal aggregator if*

$$R(f_\epsilon, \Theta) \le \min_f R(f, \Theta) + \epsilon.$$

*Proof Sketch.* The key idea here is to decompose the information structures into a linear combination of several "basic" information structures $\theta_1, \theta_2, \cdots, \theta_k$ with $\mathrm{rsupp}(\theta_i) \le 4$, where $\mathrm{rsupp}(\theta) = supp(v_1) \cup supp(v_0)$ is the support of report sets. That is, there are at most 4 possible reports in these "basic" information structures. Using the convexity of the regret function, we can prove that for any $f, \theta$, the regret $R(f, \theta)$ will be less than the linear combination of $R(f, \theta_1), \cdots, R(f, \theta_k)$. Thus we only need to solve the optimization problem among these "basic" information structures.

We then reduce the set of "basic" information structures to a constant size, which allows efficient algorithms.

*Regret.* As we cannot obtain the closed form of the optimal aggregator, we do not know the regret either. However, it is possible to estimate the regret, as stated in Lemma 4.7. Figure 2 shows an example of the regret, which is calculated by our algorithm.

**LEMMA 4.7.** *Suppose $\Theta_n$ is the information structure with $n$ truthful experts. Then the non-adversarial regret $R(\Theta_n) = c + O(1/n)$, where $c$ is a value related to $\mu, a, b$.*

*4.2.3 Partial Prior Knowledge: Adversarial Setting.* Now we consider the adversarial setting. Surprisingly, for low adversary ratio $\gamma$, the optimal aggregator has a closed form, which is also a hard sigmoid function with separation points in $\{k, n-k\}$. We state the optimal aggregator and the regret in Theorem 4.8.

**THEOREM 4.8.** *When $0 < \gamma \le \min(\frac{a}{1+a}, \frac{1-b}{2-b})$, the optimal aggregator is*

$$f^*(x) = \begin{cases} \dfrac{\mu(1-\gamma)(1-a)}{\mu(1-\gamma)(1-a) + (1-\mu)(1-2\gamma-(1-\gamma)b)} & x \le k \\[2mm] \dfrac{\mu((1-\gamma)a - \gamma)}{\mu((1-\gamma)a - \gamma) + (1-\mu)(1-\gamma)b} & x \ge n-k \\[2mm] \dfrac{x-k}{n-2k}(f(n-k) - f(k)) + f(k) & otherwise \end{cases}$$

$$(2)$$

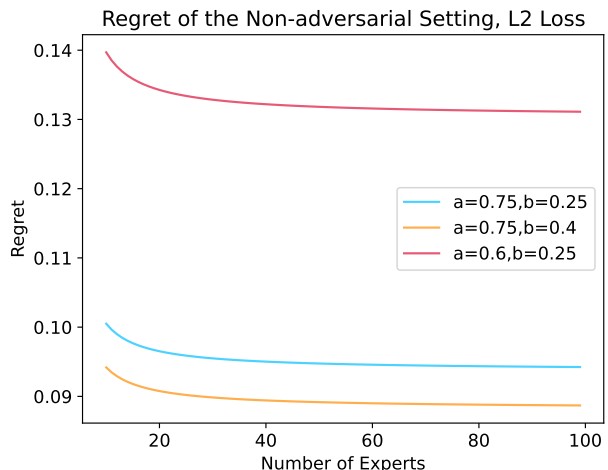

**Figure 2: Illustration for the regret under non-adversarial setting, L2 loss. We fix $\mu = 0.5$ and vary the number of experts. The parameters $a, b$ are shown in the legend.**

*Moreover, the regret $R(\Theta, \Sigma)$ is*

$$(-1 + \gamma)(-1 + \mu)\mu$$
$$* \left(b(-1 + b + 2\gamma - b\gamma) - (-1 + a + b)\left(b + a(-1 + \gamma) + \gamma - b\gamma\right)\mu\right)$$
$$/ \left(b(-1 + \gamma)(-1 + \mu) + a\mu - (1 + a)\gamma\mu\right)$$
$$/ \left(-1 + b(-1 + \gamma)(-1 + \mu) + a\mu - \gamma(-2 + \mu + a\mu)\right)$$

*Proof Sketch.* Similar to Theorem 4.2, we prove this theorem by directly constructing an equilibrium $(f^*, \theta^*, \sigma^*)$. On the one hand, $\theta^*, \sigma^*$ provide a lower bound for the regret. On the other hand, $f^*$ is the best response to $\theta^*, \sigma^*$. In addition, for a special family of aggregators $f$ including $f^*$, $(\theta^*, \sigma^*)$ is their corresponding worst case. Thus $f^*$ is optimal.

*Discussion.* If we simply substitute $\gamma = 0$ into the formula in Theorem 4.8, we can not obtain the optimal aggregator for the non-adversarial setting as Theorem 4.6 computes. Thus the adversarial setting has some essential differences from the non-adversarial setting. We will use an example to show the reason. We first consider the non-adversarial setting. Suppose there are 5 experts and we select the average aggregator $f(x) = \frac{x}{n}$. Assume $a = \frac{3}{5}$, $b = \frac{2}{5}$, and $\mu = \frac{1}{2}$.

EXAMPLE 4.9 (EXAMPLES OF THE NON-ADVERSARIAL SETTING). *Consider two information structures $\theta_1, \theta_2$.*

$$\theta_1 : \Pr_{\theta_1}[x|\omega = 1] = \begin{cases} 1/2 & x = 1, 5 \\ 0 & else \end{cases} \quad \Pr_{\theta_1}[x|\omega = 0] = \begin{cases} 1/2 & x = 0, 4 \\ 0 & else \end{cases}$$

$$\theta_2 : \Pr_{\theta_2}[x|\omega = 1] = \begin{cases} 1/2 & x = 1, 5 \\ 0 & else \end{cases} \quad \Pr_{\theta_2}[x|\omega = 0] = \begin{cases} 3/5 & x = 0 \\ 2/5 & x = 5 \\ 0 & else \end{cases}$$

In the information structure $\theta_1$, the loss of the aggregator is $\frac{1}{2}\left(\frac{1}{2}\left(1 - \frac{1}{5}\right)^2 + \frac{1}{2}\left(\frac{4}{5}\right)^2\right) = \frac{8}{25}$ and the loss of the benchmark is 0. In the information structure $\theta_2$, the loss of the aggregator is $\frac{1}{2}\left(\frac{1}{2}\left(1 - \frac{1}{5}\right)^2 + \frac{2}{5} \cdot 1^2\right) = \frac{9}{25}$, which is greater than in $\theta_1$. However, the loss of the aggregator is also greater than in $\theta_1$. Notice that the regret is the difference between the loss of the aggregator and the loss of the benchmark. Therefore, for the average aggregator, we cannot easily determine which of $\theta_1$ or $\theta_2$ has the greater regret. In fact, the worst information structure corresponding to simple averaging is a mixture of some information structures. That is why we need to solve it using an algorithm.

Surprisingly, when we add one adversary, the situation becomes simpler. We consider the information structure $\theta_1$ and adversarial strategy $\sigma(5) = \sigma(1) = 0, \sigma(4) = \sigma(0) = 1$. In this case, we obtain highest loss of the aggregator while keeping the zero loss benchmark. Thus it is easier to determine the worst information structure in the adversary setting.

## 5 Numerical Experiment

This section evaluates our aggregators in ensemble learning, where we combine multiple models' predictions to achieve higher accuracy. There exist data poisoning attacks [22] in the ensemble learning process in practice, which corresponds to the adversarial setting. The theory has already provided the worst-case analysis. The experiment focuses on the average performance with specific adversarial strategies, which reflect real-world situations and are feasible.

### 5.1 Setup

We now apply our framework to ensemble learning for image classification.

- **World State** $\omega$: it is the true class $y_j$ of the data point $d_j$.
- **Expert** $i$: it is a black-box machine learning model $M_i$, which will take the data point $d_j$ as the input and output a prediction for its class $M_i(d_j)$.
- **Signal** $s_i$: suppose we have a training dataset $\mathcal{D}$. For each model $M_i$, the signal $s_i$ is defined by its training dataset $\mathcal{D}_i \subset \mathcal{D}$.
- **Report** $x_i$: it is model $M_i$'s output class. We do not consider the confidence of models.
- **Benchmark Function** $opt$: it is defined by the best model trained using the full dataset $\mathcal{D}$.

In our experiment, we utilized the CIFAR-10 dataset, which comprises images across 10 distinct classes. To adapt this multi-class dataset for our binary signals framework, our task is to determine whether the image belongs to a special class (e.g. cat) or not. To ensure the symmetric assumption and keep the diversity of the models, we train 100 models using a consistent machine learning backbone according to Page [30]. For each model $M_i$, we construct the sub-dataset $\mathcal{D}_i$ by uniformly sampling 20000 images from the original training dataset, which contains 50000 images. We train 10 epochs by GPU RTX 3060 and CPU Intel i5-12400. The average accuracy of models is around 85%.

*Estimation For Parameters.* To apply our aggregator in Theorem 4.8, we need three important parameters, the prior $\mu$, the probability $a$ of the "yes" answer when the true label is "yes", and the probability $b$ of the "yes" answer when the true label is "no". In practice, it is impossible to know the true value due to the imperfect knowledge of the data distribution. Instead, we can estimate them by the empirical performance of models in the training set. If the training set is unbiased samples from the true distribution, the empirical estimator is also unbiased for these true parameters.

*Adversary Models.* We test two different kinds of adversarial strategies. First is the extreme strategy. That is, the adversaries always report the opposite forecast to the majority of truthful experts. Second is the random strategy, which will randomly report "yes" or "no" with equal probability.

*Aggregators.* We compare our aggregators to two benchmarks: the majority vote-outputting the answers of the majority of experts; and the averaging-outputting the ratio of models answering "yes".

## 5.2 Results

We evaluated the performance of different aggregators across a range of adversaries and Figure 3 shows our results for L2 loss. For the random adversaries, we sample 50 independent groups of adversarial experts and draw an error bar of standard deviation. More results are presented in Appendix D. Our piecewise linear aggregator (Theorem 4.8) outperforms other aggregators in any situation. Notably, the majority has a close performance to our aggregator, which means it is a good approximation in the ensemble learning setting. This is mainly due to the high accuracy of each model. When the group of adversaries is small, the averaging performs well. However, when the group becomes large, the effectiveness of averaging significantly diminishes. Thus averaging is very sensitive to the number of adversaries.

When the adversaries use the random strategy, all aggregators generally perform better. In particular, the majority vote will not be affected by the adversaries in expectation since it will not change the majority. When the number of adversaries is small, the averaging even outperforms our aggregator, as our aggregator is overly conservative in this case.

Figure 4 illustrates the accuracy of various aggregation methods. The benchmark aggregator, which represents the model trained on the full dataset, achieves approximately 98% accuracy. The k-Truncated Random Select, a randomized variant of the k-Truncated Mean, performs similarly to the majority vote, both reaching around 96% accuracy. In contrast, the Random Select, which randomly follows one model, performs poorly, especially when the number of adversaries increases.

## 6 Conclusion

We analyze the robust aggregation problem under both truthful and adversarial experts. We show that the optimal aggregator is piecewise linear across various scenarios. In particular, the truncated mean is optimal for L1 loss. We evaluate our aggregators by an ensemble learning task. For the general setting with more flexible information structures and experts' reports space, we provide

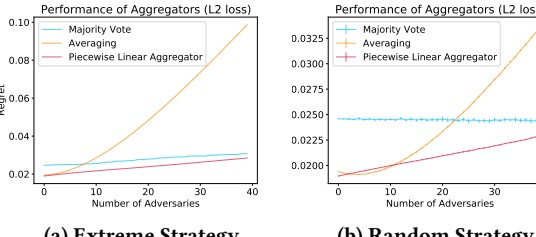

(a) Extreme Strategy   (b) Random Strategy

**Figure 3: The performance of different aggregators under fifferent adversarial strategies. The x-axis is the number of adversaries we add. The number of truthful experts is 100. The y-axis is the regret.**

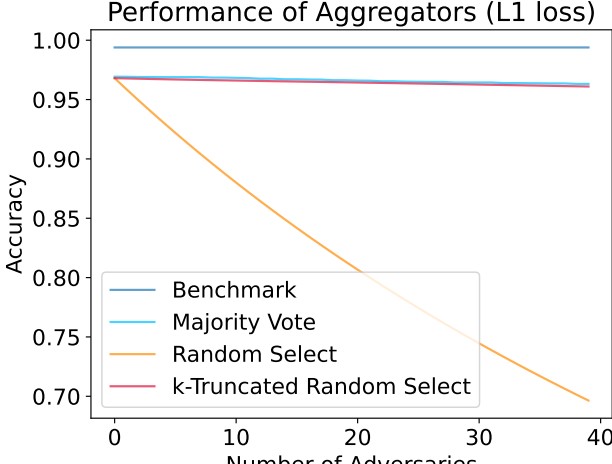

**Figure 4: The accuracy of different aggregators under extreme strategy. The x-axis is the number of adversaries we added to experts. The number of truthful experts is 100. The y-axis is the accuracy.**

some negative results that the optimal aggregator is vulnerable to adversarial experts.

For future work, it would be interesting to extend the binary world state into a multi-state case, where the adversarial strategies are more diverse. Another possible direction is exploring the performance of the truncated mean in other general information structures, such as the substitute information structure [29]. Moreover, we wonder what if we connect behavioral economics by considering other types of experts, such as experts who only report truthfully with some probability.

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

# A Auxiliary Tools

In this section, we first give some useful notations. We define two important distributions $v_1, v_0 \in \Delta_{[n]}$, which are the distributions of reports conditioning on the world state: $v_1(t) = \Pr_{\theta,\sigma}[x = t | \omega = 1]$, $v_0(t) = \Pr_{\theta,\sigma}[x = t | \omega = 0]$. Notice that $v_1, v_0$ are related to both $\theta$ and $\sigma$. Similarly, we define $u_1(t) = \Pr_{\theta}[x_T = t | \omega = 1]$, $u_0(t) = \Pr_{\theta}[x_T = t | \omega = 0]$ which are the conditional distributions of truthful experts' reports. Notice that $u_1, u_0 \in \Delta_{[n-k]}$.

The expected loss of the aggregator can be decomposed by $v_1, v_0$:

$$\mathbb{E}_{\theta,\sigma}[\ell(f(x), \omega)] = \mu E_{x \sim v_1}[\ell(f(x))] + (1 - \mu)E_{x \sim v_0}[\ell(1 - f(x))].$$

By simple calculation, we can obtain the closed-form of the benchmark $opt_\theta(x)$ under different loss function $\ell$.

**L1 loss**

$$opt_\theta(x_T) = \begin{cases} 1 & \mu u_1(x_T) \geq (1-\mu)u_0(x_T) \\ 0 & otherwise \end{cases}$$

The expected loss is $1 - \sum_x \max(\mu u_1(x), (1-\mu)u_0(x))$.

**L2 loss**

$$opt_\theta(x_T) = \frac{\mu u_1(x_T)}{\mu u_1(x_T) + (1-\mu)u_0(x_T)}.$$

The expected loss is $\sum_x \frac{\mu(1-\mu)u_1(x)u_0(x)}{\mu u_1(x) + (1-\mu)u_0(x)}$

Before considering the optimal aggregator, we first characterize all feasible information structures. Lemma A.1 fully formalizes the distribution $u_1, u_0$, i.e., the possible reports of $n - k$ truthful experts.

**LEMMA A.1 ([3]).** *For any distribution $u_1, u_0 \in \Delta_{[n-k]}^2$, there exists an information structure $\theta$ such that $u_1(t) = \Pr_\theta[x_T = t | \omega = 1]$, $u_0(t) = \Pr_\theta[x_T = t | \omega = 0]$ if an only if $\mathbb{E}_{x \sim u_1}[x] = (n-k)a$ and $\mathbb{E}_{x \sim u_0}[x] = (n-k)b$.*

The following lemma is an extension of Lemma A.1. It shows that there is also some restriction on the expectation of the distribution of inputs for adversarial experts and truthful experts.

**LEMMA A.2.** *For any information structure $\theta$ and adversarial strategy $\sigma$, the corresponding $v_1, v_0$ satisfy $(n - k)a \leq \mathbb{E}_{x \sim v_1}[x] \leq (n - k)a + k$ and $(n - k)b \leq \mathbb{E}_{x \sim v_0}[x] \leq (n - k)b + k$.*

**PROOF.** By A.1, $\mathbb{E}_{x \sim u_1}[x] = (n-k)a$ and $\mathbb{E}_{x \sim u_0}[x] = (n-k)b$. Consider $\mathbb{E}_{x \sim v_1}[x] - \mathbb{E}_{x \sim u_1}[x]$, which is the expectation of reports of adversaries. Since there are at most $k$ adversaries. Then $0 \leq \mathbb{E}_{x \sim v_1}[x] - \mathbb{E}_{x \sim u_1}[x] \leq k$. Thus $(n-k)a \leq \mathbb{E}_{x \sim v_1}[x] \leq (n-k)a+k$. Similarly, $(n-k)b \leq \mathbb{E}_{x \sim v_0}[x] \leq (n-k)b + k$. □

Now we give a useful lemma for our main theorems.

Definition A.3 (A Bad Information Structure). *If $k < (n - k)a$, $(n - k)b < n - 2k$, we define a bad information structure $\theta_b, \sigma_b$ such that*

$$\Pr_{\theta_b}[x = n - k | \omega = 1] = \frac{(n - k)a - k}{n - 2k}$$

$$\Pr_{\theta_b}[x = k | \omega = 1] = \frac{n - k - (n - k)a}{n - 2k}$$

$$\Pr_{\theta_b}[x = n - 2k | \omega = 0] = \frac{(n - k)b}{n - 2k}$$

$$\Pr_{\theta_b}[x = 0 | \omega = 0] = \frac{n - 2k - (n - k)b}{n - 2k}$$

*and*

$$\sigma_b(x) = \begin{cases} k & x = 0, n - k \\ 0 & otherwise \end{cases} \tag{3}$$

Lemma A.4. *If the aggregator $f$ satisfies the following conditions:*
- *non-decreasing*
- *$\ell(f(x), 0)$ is convex in $[k, n - k]$*
- *$\ell(f(x), 1)$ is convex in $[k, n - k]$*
- *$f(x)$ is constant in $[0, k]$ and $[n - k, k]$.*

*Then $R(f, \Theta, \Sigma) = R(f, \theta_b, \sigma_b)$.*

Proof.

$$\mathbb{E}_{\theta,\sigma}[\ell(f)] = \mu \sum_x \mathbb{E}_\sigma[v_1(x)\ell(f(x + \sigma(x)), 1)]$$

$$+ (1 - \mu) \sum_x \mathbb{E}_\sigma[v_0(x)\ell(f(x + \sigma(x)), 0)]$$

$$\leq \mu \sum_x u_1(x)\ell(f(x), 1) + (1 - \mu) \sum_x u_0(x)\ell(f(x + k), 0)$$
$$(f \text{ is non-decreasing})$$

$$= \mu \sum_{n - k \geq x \geq k} u_1(x)(\alpha(x)k$$

$$+ (1 - \alpha(x))(n - k)\ell(f(x), 1)$$

$$+ \mu \sum_{x < k} u_1(x)\ell(f(x), 1) \qquad (\alpha(x) = \frac{n - k - x}{n - 2k})$$

$$+ (1 - \mu) \sum_{x \leq n - 2k} u_0(x)(\beta(x)0$$

$$+ (1 - \beta(x))(n - 2k))\ell(f(x + k), 0)$$

$$+ (1 - \mu) \sum_{x > n - 2k} u_0'(x)\ell(f(x), 0) \qquad (\beta(x) = \frac{n - 2k - x}{n - 2k})$$

$$\leq \mu \sum_{x \in \{k, n-k\}} u_1'(x)\ell(f(x), 1) + \mu \sum_{x < k} u_1'(x)\ell(f(x), 1)$$
$$(\ell(f(x), 1) \text{ is convex in } [k, n - k])$$

$$+ (1 - \mu) \sum_{x \in \{0, n-2k\}} u_0'(x)\ell(f(x + k), 0)$$

$$+ (1 - \mu) \sum_{x > n - 2k} u_0'(x)\ell(f(x + k), 0)$$
$$(\ell(f(x), 0) \text{ is convex in } [k, n - k])$$

where $u_1'(x) = u_1(x)$ for any $x < k$, $u_0'(x) = u_0(x)$ for any $x > n - 2k$

$$u_1'(k) = \sum_{k \leq x \leq n-k} \alpha(x)u_1(x),$$

$$u_1'(n - k) = \sum_{k \leq x \leq n-k} (1 - \alpha(x))u_1(x).$$

$$u_0'(0) = \sum_{0 \leq x \leq n-2k} \beta(x)u_0(x),$$

$$u_1'(n - 2k) = \sum_{0 \leq x \leq n-2k} (1 - \beta(x))u_0(x).$$

By simple calculation we have $\sum_x u_1'(x) = 1$, $\sum_x xu_1'(x) = (n - k)a$ and $\sum_x u_0'(x) = 1$, $\sum_x xu_0'(x) = (n - k)b$.

Now we calculate the maximum of

$$\sum_{x \in \{k, n\}} u_1'(x)\ell(f(x), 1) + \sum_{x < k} u_1'(x)\ell(f(x), 1)$$

subject to

$$\sum_x u_1'(x) = 1$$

$$\sum_x xu_1'(x) = (n - k)a$$

Notice that $f(x)$ is constant when $x \leq k$. For any $x \leq k$, we write $\alpha(x) = \frac{n - x}{n - 2k}$, then

$$u_1'(x)\ell(f(x), 1) = u_1'(x)(\alpha(x)\ell(f(x), 1) + (1 - \alpha(x))\ell(f(x), 1))$$

$$\leq u_1'(x)(\alpha(x)\ell(f(k), 1) + (1 - \alpha(x))\ell(f(n - k), 1)$$
$$(f(x) = f(k) \leq f(n - k) \text{ for } x \leq k)$$

Thus we can replace any $x < k$ with the linear combination of $k, n - k$ until there are only reports $k, n - k$ left. That is,

$$\sum_{x \in \{k, n-k\}} u_1'(x)\ell(f(x), 1) + \sum_{x < k} u_1'(x)\ell(f(x), 1)$$

$$\leq \sum_{x = \{k, n-k\}} u_1''(x)\ell(f(x), 1)$$

where

$$\sum_x u_1''(x) = 1$$

$$\sum_x xu_1''(x) = (n - k)a$$

Similarly, we have

$$\sum_{x \in \{0, n-2k\}} u_0'(x)\ell(f(x), 0) + (1 - \mu) \sum_{x > n-k} u_0'(x)\ell(f(x), 0)$$

$$\leq \sum_{x = \{0, n-2k\}} u_0''(x)\ell(f(x), 0)$$

where

$$\sum_x u_0''(x) = 1$$

$$\sum_x xu_0''(x) = (n - k)b$$

In fact, $u_1'', u_0''$ is the same as $\theta_b$. Thus $R(f, \theta, \sigma) \leq R(f, \theta_b, \sigma_b)$ for any $\theta, \sigma$, which completes our proof. □

## B Omitted Proofs in Section 4

### B.1 Proof of Theorem 4.2

On the one hand, it is easy to verify that $k$-truncated mean satisfies the condition in Lemma A.4. Thus $R(f^*, \theta_b, \sigma_b) = R(f^*, \Theta, \Sigma)$

On the other hand, we have

$$\Pr_{\theta_b, \sigma_b}[x|\omega = 1] = \begin{cases} \dfrac{(n-k)a-k}{n-2k} & x = n-k \\ \dfrac{(n-k)(1-a)}{n-2k} & x = k \\ 0 & otherwise \end{cases} \quad (4)$$

and

$$\Pr_{\theta_b, \sigma_b}[x|\omega = 0] = \begin{cases} \dfrac{(n-k)b}{n-2k} & x = n-2k \\ \dfrac{n-2k-(n-k)b}{n-2k} & x = 0 \\ 0 & otherwise \end{cases} \quad (5)$$

Since $\mu u_1(n-k) \geq (1-\mu)u_0(n-2k)$ and $\mu u_1(k) \leq (1-\mu)u_0(0)$, the $opt_{\theta_b, \sigma_b}(n-k) = 1, opt_{\theta_b, \sigma_b}(k) = 0$.

Thus $R(f, \Theta, \Sigma) \geq R(f, \theta_b, \sigma_b) \geq R(f^*, \theta_b, \sigma_b)$.

Combine these two claims we complete our proof.

### B.2 Proof of Lemma 4.3

Proof. On the one hand, by Lemma A.2, $(n-k)a \leq \mathbb{E}_{x \sim v_1}[x]$ and $\mathbb{E}_{x \sim v_0}[x] \leq (n-k)b + k$, then

$$R(f_t^*) = \mu(1 - \mathbb{E}_{x \sim v_1} f_t^*(x)) + (1-\mu)\mathbb{E}_{x \sim v_0} f_t^*(x)$$

$$\leq \mu(1 - \frac{(n-k)a}{n-2k'}) + (1-\mu)\frac{(n-k)b+k}{n-2k'}$$

$$= \frac{k'-k+(n-k)(1-(1-\mu)(1-b)-\mu a)}{n-2k'}$$

On the other hand, let

$$\Pr_{\theta_b, \sigma_b}[x|\omega = 1] = \begin{cases} \dfrac{(n-k)a-k}{n-2k} & x = n-k \\ \dfrac{(n-k)(1-a)}{n-2k} & x = k \\ 0 & otherwise \end{cases} \quad (6)$$

and

$$\Pr_{\theta_b, \sigma_b}[x|\omega = 0] = \begin{cases} \dfrac{(n-k)b}{n-2k} & x = n-2k \\ \dfrac{n-2k-(n-k)b}{n-2k} & x = 0 \\ 0 & otherwise \end{cases} \quad (7)$$

We have $R(f_t^*, \theta, \sigma) = \frac{k'-k+(n-k)(1-(1-\mu)(1-b)-\mu a)}{n-2k'}$, which completes our proof.

□

### B.3 Proofs of Lemma 4.4 and Lemma 4.5

Proof of Lemma 4.4. We first consider the non-adversarial setting. On the one hand, since $\ell(1/2) = 1/4$, for any $\theta, \sigma$,

$$R(f^*, \theta, \sigma) \geq \Pr_{\theta}[\omega = 1]\ell(1/2) + \Pr_{\theta}[\omega = 0]\ell(1/2)$$

$$= 1/4$$

Thus $R(f^*, \Theta, \Sigma) = 1/4$.

On the other hand, we select $x_1 = 0, x_2 = 1, x_3 = n-1, x_4 = n$, and $\mu = 0.5$. Consider the mixture of two information structures with uniform distribution. In the first one, $supp(v_1) = \{x_1, x_4\}, supp(v_0) = \{x_2, x_3\}$. In the second one, $supp(v_1) = \{x_2, x_3\}, supp(v_0) = \{x_1, x_4\}$. Let $b = 2/n$, consider a sequence $\{a_t\}$ such that $\lim_{t \to \infty} a_t = b$ and $a_t > b$.

By this sequence of parameter $a$ we construct a sequence of mixed information structures $\theta_t$. We will find that $\lim_{t \to \infty} R(\theta_t) = 1/4$ since the Bayesian posterior of $\theta_t$ is $f(x_1) = f(x_1) = f(x_3) = f(x_4) = 1/2$. Thus $R(f, \Theta, \Sigma) \geq 1/4$ for any $f$.

For the adversarial setting, the random guess will still obtain regret $1/4$. the regret of other aggregators will not decrease. Thus we complete our proof for both adversarial and non-adversarial setting.

□

Proof of Lemma 4.5. Similarly, we first consider the non-adversarial setting. On the one hand,

$$R(f^*, \theta, \sigma) \geq \Pr_{\theta}[\omega = 1]\ell(\mu) + \Pr_{\theta}[\omega = 0]\ell(1-\mu)$$

$$= \mu(1-\mu)$$

Again, we select $x_1 = 0, x_2 = 1, x_3 = n-1, x_4 = n$. Consider the mixture of two information structures. In the first one, $supp(v_1) = \{x_1, x_4\}, supp(v_0) = \{x_2, x_3\}$. In the second one, $supp(v_1) = \{x_2, x_3\}, supp(v_0) = \{x_1, x_4\}$. Using the same argument in Lemma 4.4 we will obtain that for any $f$, $R(f, \theta, \sigma) \geq \mu(1-\mu)$. Thus $f^*$ is the optimal aggregator.

□

### B.4 Proof of Theorem 4.6

We prove this theorem in several steps. First, similar to the flow in Arieli et al. [2], we do the basic dimension reduction on $\Theta$, which allows us to consider information structures with at most 4 possible reports (Lemma B.1).

Lemma B.1. Let $\Theta_4 = \{\theta \in \Theta | supp(u_1) \leq 2, supp(u_0) \leq 2\}$. Then for any $f$, we have $R(f, \Theta, \Sigma) = R(f, \Theta_4, \Sigma)$.

Proof. Consider the distribution vector $u_1$, it should satisfy the following two linear constraints:

$$\sum_x u_1(x) = 1$$

$$\sum_x x u_1(x) = na$$

By the basic theorem of linear programming, we can decompose $u_1$ with a linear combination of some extreme distribution vectors. That is, there exists $t_1, \cdots, t_m$ and $\lambda_1, \cdots, \lambda_m$ such that

$$u_1(x) = \sum_i \lambda_i t_i(x)$$

$$\sum_x t_i(x) = 1$$

$$\sum_x x t_i(x) = na$$

The extreme point here means that $supp(t_i) \leq 2$ for any $i$. Now fix $u_0$, we can create a new information structure $\theta_i$ by setting the $t_i(x) = \Pr_{\theta_i}[X = x | \omega = 1]$ and $u_0(x) = \Pr_{\theta_i}[X = x | \omega = 0]$.

Since the benchmark aggregator $\frac{u_1(x)u_0(x)}{u_1(x)+u_0(x)}$ is concave with $u_1(x)$ given $u_0(x)$, and $\mathbb{E}_\theta[\ell(f)]$ is linear in $u_1$ given $f$. Thus for any $f$,

$$R(f, \theta) \leq \sum_i \lambda_i R(f, \theta_i) \leq \max_i R(f, \theta_i).$$

Then following the same argument in $u_0$ we obtain that we can decompose $u_0$ with some extreme vectors with support space less than 2, which completes our proof for $R(f, \Theta) = R(f, \Theta_4)$.

□

However, we still have around $\binom{n}{4}$ possible information structures. Furthermore, we show that we can only consider those "extreme information structures" with extreme report support space (Lemma B.2). "Extreme" here means reports in $\{0, 1, n-1, n\}$. Thus the meaningful information structures are reduced to constant size.

LEMMA B.2. Let $\Theta_e = \{\theta \in \Theta_4 | supp(u_1) \subset \{0, 1, n-1, n\}, supp(u_0) \subset \{0, 1, n-1, n\}\}$. We have $R(\Theta, \Sigma) = R(\Theta_e, \Sigma)$.

PROOF. Consider the optimal aggregator $f^* = \arg\min_f R(f, \Theta_e)$. We extend $f^*$ by linear interpolation in the non-extreme input:

$$f(x) = \begin{cases} f^*(x) & x = \{0, 1, n-1, n\} \\ \dfrac{f^*(n-1) - f^*(1)}{n-2} * (x-1) & otherwise \end{cases} \quad (8)$$

Then for any $2 \leq x \leq n-2$, we can write $x$ as linear combination of 1 and $n-1$, $x = \alpha(x) * 1 + (1 - \alpha(x)) * (n-1)$ where $\alpha(x) = \frac{n-1-x}{n-2} \in [0, 1]$. Since both $(1 - f(x))^2$ and $f(x)^2$ is convex in $[1, n-1]$,

$$\mathbb{E}_\theta[\ell(f)]$$

$$= \mu \sum_x u_1(x)(1 - f(x))^2 + (1 - \mu) \sum_x u_0(x) f(x)^2$$

$$\leq \mu \sum_{x \in \{0,1,n-1,n\}} u_1(x)(1 - f(x))^2$$

$$+ \mu \sum_{x \notin \{0,1,n-1,n\}} u_1(x) \left( \alpha(x)(1 - f(1))^2 + (1 - \alpha(x))(1 - f(n-1))^2 \right)$$

$$+ (1 - \mu) \sum_{x \in \{0,1,n-1,n\}} u_0(x) f(x)^2$$

$$+ (1 - \mu) \sum_{x \notin \{0,1,n-1,n\}} u_0(x) \left( \alpha(x) f(1)^2 + (1 - \alpha(x)) f(n-1)^2 \right)$$

(Convexity)

$$= \mu \sum_{x \in \{0,1,n-1,n\}} v_1'(x)(1 - f(x))^2$$

$$+ (1 - \mu) \sum_{x \in \{0,1,n-1,n\}} v_0'(x) f(x)^2$$

where $u_1'(0) = u_1(0), u_1'(n) = u_1(n)$ and

$$u_1'(1) = u_1(1) + \sum_{2 \leq x \leq n-2} \alpha(x) u_1(x)$$

$$u_1'(n-1) = u_1(1) + \sum_{2 \leq x \leq n-2} (1 - \alpha(x)) u_1(x)$$

Thus we have

$$\sum_x u_1'(x)$$

$$= u_1'(0) + u_1'(1) + u_1'(n-1) + u_1'(n)$$

$$= u_1(0) + u_1(1) + u_1(n-1) + u_1(n)$$

$$+ \sum_{2 \leq x \leq n-2} \alpha(x) u_1(x) + (1 - \alpha) u_1(x)$$

$$= u_1(0) + u_1(1) + u_1(n-1) + u_1(n) + \sum_{2 \leq x \leq n-2} u_1(x)$$

$$= 1$$

and

$$\sum_x x u_1'(x)$$

$$= u_1'(1) + (n-1) u_1'(n-1) + n u_1'(n)$$

$$= u_1(1) + (n-1) u_1(n-1) + n u_1(n)$$

$$+ \sum_{2 \leq x \leq n-2} \alpha(x) u_1(x) + (1 - \alpha)(n-1) u_1(x)$$

$$= u_1(0) + u_1(1) + (n-1) u_1(n-1) + n u_1(n) + \sum_{2 \leq x \leq n-2} x u_1(x)$$

$$= a$$

Similarly we have $\sum_x u_0'(x) = 1, \sum_x x u_0'(x) = b$. We can create a new information structure $\theta'$ by setting $u_1'(x) = \Pr_{\theta_i}[X = x | \omega = 1]$ and $u_0'(x) = \Pr_{\theta_i}[X = x | \omega = 0]$. Notice that $\theta' \in \Theta_e$. Thus for any $\theta \in \Theta$, we have $R(f, \theta) \leq R(f, \theta')$. So

$$R(\Theta) = \min_{f'} R(f', \Theta) \leq R(f, \Theta) \leq R(f, \Theta_e) = R(f^*, \Theta_e) = R(\Theta_e)$$

.

However, since $\Theta_e \subset \Theta$, $R(\Theta_e) \leq R(\Theta)$. So we obtain that $R(\Theta) = R(\Theta_e)$.

□

Combining these two lemmas we only need to consider information structures with at most 4 possible reports and the reports are in $\{0, 1, n-1, n\}$. In addition, $supp(u_1) \leq 2, supp(u_0) \leq 2$. There are at most 16 possible information structures and there exists an FPTAS for solving the finite number of information structures by Guo et al. [16].

LEMMA B.3 ([16]). Suppose $|\Theta| = n$, There exists an FPTAS which can find the $\epsilon$-optimal aggregator in $O(n/\epsilon)$.

If we apply the FPTAS in $\Theta_e$ we will find the $\epsilon$-optimal aggregator in $O(1/\epsilon)$, which completes our proof.

## B.5 Proof of Lemma 4.7

Suppose $R_n$ is the regret when the number of experts is $n$. We want to estimate the difference $R_n - R_{n+1}$. Let $f_1 = \arg\min_f \max_{\theta \in \Theta_{n+1}} R(f, \theta)$. As we prove in Lemma B.2, we only need to consider point $0, 1, n-1, n$. Then for any information structure $\theta_1$ in $\Theta_{n+1}$, we map it to another information structure $\theta_2 \in \Theta_n$ by the following rules. If $\theta_1$ contains report $0, 1$, let $\theta_2$ contains report $0, 1$; if $\theta_1$ contains report $n$ or $n+1$, let $\theta_2$ contains report $n$ or $n-1$. Then the joint distribution is determined by the report space. Now we construct an aggregator $f_2$:

$$f_2(x) = \begin{cases} f_1(x) & x = 0, 1 \\ f_1(x-1) & x = n, n-1 \end{cases} \quad (9)$$

Then we have

$$R(f_2, \theta_2) - R(f_1, \theta_1) = \mathbb{E}_{x \sim \theta_2}[(f(x) - opt_{\theta_2}(x))^2]$$
$$- \mathbb{E}_{x \sim \theta_1}[(f(x) - opt_{\theta_1}(x))^2]$$
$$= \sum_x \Pr_{\theta_2}[x](f(x) - opt_{\theta_2}(x))^2$$
$$- \Pr_{\theta_1}[x](f(x) - opt_{\theta_1}(x))^2$$
$$\leq \sum_x \left| \Pr_{\theta_2}[x] - \Pr_{\theta_1}[x] \right|$$
$$((f(x) - opt_{\theta_2}(x))^2 \leq 1)$$
$$\leq O\left(\frac{1}{n(n+1)}\right)$$

Thus $R(\Theta_n) \leq R(f_2, \Theta_{n+1}) \leq R(f_1, \Theta_n) + O\left(\frac{1}{n(n+1)}\right) = R(\Theta_{n+1}) + O\left(\frac{1}{n(n+1)}\right)$.

Add up all $n$ we have $R(\Theta_n) \leq \sum_k O\left(\frac{1}{k(k+1)}\right) = c + O(\frac{1}{n})$

## B.6 Proof of Theorem 4.8

On the one hand, it is easy to verify that $k$-truncated mean satisfies the condition in Lemma A.4. Thus $R(f^*, \theta_b, \sigma_b) = R(f^*, \Theta, \Sigma)$

On the other hand, we have

$$\Pr_{\theta_b, \sigma_b}[X = x | \omega = 1] = \begin{cases} \dfrac{(n-k)a-k}{n-2k} & x = n-k \\ \dfrac{(n-k)(1-a)}{n-2k} & x = k \\ 0 & otherwise \end{cases} \quad (10)$$

and

$$\Pr_{\theta_b, \sigma_b}[X = x | \omega = 0] = \begin{cases} \dfrac{(n-k)b}{n-2k} & x = n-2k \\ \dfrac{n-2k-(n-k)b}{n-2k} & x = 0 \\ 0 & otherwise \end{cases} \quad (11)$$

By simple calculate we find that $opt_{\theta_b, \sigma_b}(x) = \Pr_{\theta_b, \sigma_b}[\omega = 1 | x] = f^*$.

Thus $R(f, \Theta, \Sigma) \geq R(f, \theta_b, \sigma_b) \geq R(f^*, \theta_b, \sigma_b)$.

Combine these two claims we complete our proof.

## C Hard Aggregators

In this section, we discuss a special family of aggregators-the hard aggregators that can randomly output a decision in $\{0, 1\}$. In this case, L1 loss and L2 loss are equivalent. We prove that the $k$-ignorance random dictator is optimal. It echos the results in [3] that the random dictator is optimal for the non-adversarial setting.

DEFINITION C.1 ($k$-IGNORANCE RANDOM DICTATOR). *We call $f$ is $k$-ignorance random dictator if*

$$f(x) = \begin{cases} 1 & x \geq n-k \\ 0 & x \leq k \\ report\ 1\ with\ probability\ \dfrac{x-k}{n-2k} & otherwise \end{cases} \quad (12)$$

THEOREM C.2. *When $\gamma \leq \min\left(\frac{a\mu-(1-\mu)b}{\mu+a\mu-(1-\mu)b}, \frac{a}{1+a}, \frac{1-b}{2-b}\right)$, the $k$-ignorance random dictator is optimal for both L1 and L2 loss. Moreover, the regret*

$$R(\Theta, \Sigma) = \mu + \frac{(1-\mu)((1-\gamma)b+\gamma) - \mu(1-\gamma)a}{1-2\gamma}.$$

PROOF. Notice that for both L1 and L2 loss

$$\ell(y, \omega) = \begin{cases} 1 & y = \omega \\ 0 & y \neq \omega \end{cases} \quad (13)$$

As the soft aggregators include hard aggregators, we only need to prove that the $k$-ignorance random dictator $f_1$ has the same regret as the $k$-truncated mean $f_2$ under L1 loss. In fact,

$$R(f_1, \Theta, \Sigma) = \sup_{\theta \in \theta, \sigma \in \Sigma} \mathbb{E}_{f_1, \theta, \sigma}[\ell_1(f(x), \omega)] - \mathbb{E}_\theta[\ell_1(opt_\theta(x_T), \omega)]$$
$$= \sup_{\theta \in \theta, \sigma \in \Sigma} \mathbb{E}_{\theta, \sigma}(\Pr[f_1(x) = 1]\ell_1(1, \omega)$$
$$+ \Pr[f_1(x) = 0]\ell_1(0, \omega)) - \mathbb{E}_\theta[\ell_1(opt_\theta(x_T), \omega)]$$
$$= \sup_{\theta \in \theta, \sigma \in \Sigma} \mathbb{E}_{\theta, \sigma}(f_2(x)\ell_1(1, \omega) + (1 - f_2(x))\ell_1(0, \omega))$$
$$- \mathbb{E}_\theta[\ell_1(opt_\theta(x_T), \omega)]$$
$$= \sup_{\theta \in \theta, \sigma \in \Sigma} \mathbb{E}_{\theta, \sigma}\ell_1(f_2(x), \omega) + (1 - f_2(x))\ell_1(0, \omega))$$
$$- \mathbb{E}_\theta[\ell_1(opt_\theta(x_T), \omega)]$$
$$= R(f_2, \Theta, \Sigma)$$

Thus the $k$-ignorance random dictator is optimal. □

## D Numerical Experiment

Figure 5 shows the results under L1 loss. The optimal aggregator and majority vote have close performance. It also reflects our discussion that L1 loss will encourage the aggregators to output a decision, which is beneficial for the majority vote.

## E Extension

In this section, we extend our adversarial information aggregation problem into a general case. In the general setting, we can capture a non-binary state space $\Omega$, other families of information structures $\Theta$, and other kinds of reports $x_i$ such as the posterior forecast $x_i = \Pr_\theta[\omega|s_i]$.

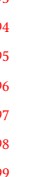
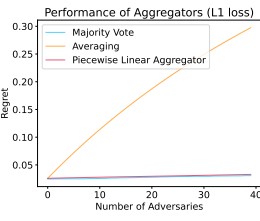
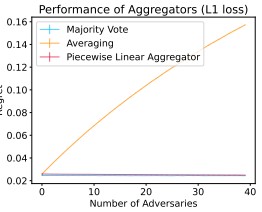

**(a) Extreme Strategy**     **(b) Random Strategy**

**Figure 5: The performance of different aggregators under different adversarial strategies. The x-axis is the number of adversaries we added to experts. The number of truthful experts is 100. The y-axis is the regret.**

### E.1 Problem Statement

Suppose the world has a state $\omega \in \Omega$, $|\Omega| = m$. There are $n$ experts and each expert $i$ receives a private signal $s_i$ in a signal space $\mathcal{S}_i$. Let $\mathcal{S} = \mathcal{S}_1 \times \mathcal{S}_2 \times ... \times \mathcal{S}_n$. They are asked to give a report from a feasible choice set $X$ according to their private signals. We denote the joint distribution, or information structure, over the state and signals by $\theta \in \Delta_{\Omega \times \mathcal{S}}$. For simplicity, we denote $\boldsymbol{x}_T = (x_t)_{t \in T}$ as the sub-vector for any vector $\boldsymbol{x}$ and index set $T$.

We assume the experts are either truthful or adversarial. Let $T$ denote the set of truthful experts who will give their best report truthfully. $A$ is the set of adversarial experts who will collude and follow a randomized strategy $\sigma_A : \mathcal{S}_A \to \Delta_{X^k}$ depending on their private signals, where $k = \gamma n$ is the number of adversarial experts. The family of all possible strategies is $\Sigma$. We assume $\gamma < 1/2$ such that there exist at least half truthful experts, otherwise no aggregators can be effective.

Notice that the ability of adversarial experts can be modeled by their private signals. For example, the omniscient adversarial experts can know truthful experts' signals $\mathcal{S}_R$ and the world's true state $\omega$. The ignorant adversarial experts are non-informative, i.e. receiving nothing.

The aggregator is an anonymous function $f(\cdot) \in \mathcal{F}$ which maps $\boldsymbol{x} \in X^n$ to a distribution $\boldsymbol{y} \in \Delta_\Omega$ over the world state. We define a loss function $\ell(\boldsymbol{y}, \omega) : \Delta_\Omega \times \Omega \to R^+$, indicating the loss suffered by the aggregator when the aggregator's predicted distribution of the state is $\boldsymbol{y}$ and the true state is $\omega$. The expected loss is $\mathbb{E}_{\theta,\sigma}[\ell(f(\boldsymbol{x}), \omega)]$. We assume $\ell$ is symmetric and convex for any state, which means we can abbreviate $\ell(\boldsymbol{y}, \omega)$ by $\ell(y_\omega)$. Without loss of generality, we assume $\ell(0) = 1, \ell(1) = 0$ and $\ell(\cdot)$ is decreasing. In particular, we consider the L1 loss (or absolute loss) $\ell_1(y) = 1 - y$ and the L2 loss (or square loss), $\ell_2(y) = (1 - y)^2$. We define a benchmark function, that gives the optimal result given the joint distribution and truthful experts' reports:

$$opt_\theta(\boldsymbol{x}_T) = \arg\min_{f' \in \mathcal{F}} \mathbb{E}_\theta[\ell(f'(\boldsymbol{x}_T)_\omega)]$$

to minimize the expected loss.

Under the L1 loss, $opt_\theta(\boldsymbol{x}_T)_\omega = \mathbb{1}(\omega = \arg\max_\omega \Pr_\theta[\omega|\boldsymbol{x}_T])$, which is the maximum likelihood. Under the L2 loss, $opt_\theta(\boldsymbol{x}_T)_\omega = \Pr_\theta[\omega|\boldsymbol{x}_T]$, which is the Bayesian posterior.

*Regret Robust Paradigm.* Given a family of joint distributions $\Theta$ and a family of strategies $\Sigma$, a set of aggregators $\mathcal{F}$, we aim to minimize the expected loss in the worst information structure. That is, we want to find an optimal function $f^*$ to solve the following min-max problem:

$$R(\Theta, \Sigma) = \inf_{f \in \mathcal{F}} \sup_{\theta \in \theta, \sigma \in \Sigma} \mathbb{E}_{\theta,\sigma}[\ell(f(\boldsymbol{x})_\omega)] - \mathbb{E}_\theta[\ell(opt_\theta(\boldsymbol{x}_T)_\omega)].$$

Again we define $R(f, \theta, \sigma) = \mathbb{E}_{\theta,\sigma}[\ell(f(\boldsymbol{x})_\omega)] - \mathbb{E}_\theta[\ell(opt_\theta(\boldsymbol{x}_T)_\omega)]$ and $R(f, \Theta, \Sigma) = \sup_{\theta \in \Theta, \sigma \in \Sigma} R(f, \theta, \sigma)$ for short.

### E.2 Negative Results For General Model

We first provide an auxiliary lemma to characterize the behavior of adversaries.

LEMMA E.1. *Assume $\Sigma_p = \{\sigma | \sigma \in \Sigma$ and $\sigma$ is a pure strategy$\}$. Then $R(f, \Theta, \Sigma) = R(f, \Theta, \Sigma_p)$ for any $f$ and $\Theta$.*

PROOF. On the one hand, $\Sigma_p \in \Sigma$, so $R(f, \theta, \Sigma) \geq R(f, \theta, \Sigma_p)$. On the other hand, for any $f, \theta, \sigma$,

$$R(f, \theta, \sigma) = \mathbb{E}_\theta \mathbb{E}_\sigma[\ell(f(\boldsymbol{x}_T, \sigma(\boldsymbol{s}_A)), \omega)] - \mathbb{E}_\theta[\ell(opt_\theta(\boldsymbol{x}_T)_\omega)]$$
$$\leq \mathbb{E}_\theta[\max_{\boldsymbol{x}'_A} \ell(f(\boldsymbol{x}_T, \boldsymbol{x}'_A(\boldsymbol{s}_A)), \omega)] - \mathbb{E}_\theta[\ell(opt_\theta(\boldsymbol{x}_T)_\omega)]$$

Fix the aggregator $f$ and distribution $\theta$, we can let $\sigma'(\boldsymbol{s}_A) = \arg\max_{\boldsymbol{x}'_A} \mathbb{E}_\theta[\ell(f(\boldsymbol{x}_T, \boldsymbol{x}'_A(\boldsymbol{s}_A)), \omega)]$ (We arbitrarily select one $\boldsymbol{x}'_A$ when there is multiple choices). Since $\sigma'$ is a pure strategy, $\sigma' \in \Sigma^d$. For any $\sigma \in \Sigma$, $R(f, \theta, \sigma') \geq R(f, \theta, \sigma)$. Thus $R(f, \theta, \Sigma) \leq R(f, \theta, \Sigma_p)$, which completes our proof. □

### E.3 Extension to Multi-State Case

For the multi-state case: $|\Omega| > 2$, we can also define the sensitive parameter by enumerating each state:

DEFINITION E.2 (SENSITIVE PARAMETER, MULTIPLE). *When $|\Omega| > 2$, for any benchmark function opt, the sensitive parameter is defined by*

$$S(opt, k) = \max_{\theta, \theta' \in \Theta, d(\boldsymbol{x}_T, \boldsymbol{x}'_T) \leq k, \omega \in \Omega} |opt_\theta(\boldsymbol{x}_T)_\omega - opt_{\theta'}(\boldsymbol{x}'_T)_\omega|.$$

First, we prove that for the general model, aggregators are vulnerable to adversaries. A direct observation is that fixing the number of experts $n$ while increasing the number of adversaries $k$ will not decrease the regret $R(\Theta, \Sigma)$. Since the new adversaries can always pretend to be truthful experts. In most cases, when $\gamma \approx 1/2$, it is impossible to design any informative aggregator since an adversarial expert can report opposite views to another truthful expert.

One natural question is how many adversaries we need to attack the aggregator. For this question, we provide a negative result. The following lemma shows that for a large family of information structures, a few adversaries are enough to fool the aggregator. Surprisingly, the number of adversaries is independent of the number of truthful experts but grows linearly with the number of states. In special, in the binary state setting, one adversary is enough to completely fool the aggregator.

LEMMA E.3. *Assume the truthful experts are asked to report the posterior $x_i(\omega) = \Pr_\theta[\omega|s_i]$. We define the fully informative expert*

*who always knows the true state and the non-informative expert who only knows the prior. If $\Theta$ includes all information structures consisting of these two types of experts, then for $k \geq m-1$, the optimal aggregator is the uniform prediction. That is, $f^* = (\frac{1}{m}, \cdots, \frac{1}{m})$ and $R(\theta, \sigma) = \ell(\frac{1}{m})$.*

PROOF. We select one fully informative expert, which means she will report a unit vector $e_i = (0, \cdots, 1, 0, \cdots, 0)$ with $e_i = 1$ and $e_j = 0$ for any $j \neq i$. Other experts are non-informative and will always report the uniform prior $(\frac{1}{m}, \cdots, \frac{1}{m})$. Then let the $m - 1$ adversaries report other unit vectors $e_j$ for $i \neq j$. Other adversaries also report the uniform prior. In that case, the aggregator will always see the same reports $x^0$, and the benchmark can follow the informative expert and suffer zero loss. Since $\ell$ is convex and $\sum_{\omega \in \Omega} f(x^0)_\omega = 1$, then for any $f, R(f, \theta, \sigma) = \sum_{\omega \in \Omega} \ell(f(x^0)_\omega) \geq \ell(\frac{1}{m})$. Thus $R(\Theta, \Sigma) \geq \ell(\frac{1}{m})$.

On the other hand, when $f^* = (\frac{1}{m}, \cdots, \frac{1}{m})$,

$$R(f^*, \theta, g) = \sum_{\omega \in \Omega} \Pr[\omega] \ell(\frac{1}{m}) = \ell(\frac{1}{m})$$

for any $\theta, \sigma$. So the uniform prediction is the optimal aggregator and $R(\Theta, \Sigma) = \ell(\frac{1}{m})$.

□

The uniform prediction means that we cannot obtain any additional information from reports. It is almost impossible in the non-adversarial setting. We provide a common setting as an example.

EXAMPLE E.4 (CONDITIONALLY INDEPENDENT SETTING). *Conditionally independent setting $\Theta^{ci}$ means that every expert receives independent signals conditioning on the world state $\omega$. Formally, for each $\theta \in \Theta^{ci}$, for all $s_i \in S_i, \omega \in \Omega = \{0, 1\}, \Pr_\theta[s_1, \cdots, s_n | \omega] = \Pi_i \Pr_\theta[s_i | \omega]$.*

COROLLARY E.5. *In the conditionally independent setting, if we select $\ell$ as the L2 loss, then for any $n$ and $k \geq 1, R = 1/4$.*

Notice that in the non-adversarial setting, when $n \to \infty, R \to 1/4$ [2], which is a strict condition. However, in the adversarial setting, we only use one adversary to obtain the same bad regret.

## E.4 Estimate the Regret $R(\Theta, \Sigma)$

We have provided a negative result when there are enough adversarial experts. In this section, we want to extend our result regarding the regret to any number of adversarial experts setting. It will give us a further understanding of the effect of adversaries.

Intuitively, when there exist important experts, it is easier to disturb the aggregator because adversaries can always imitate an important expert but hold the opposite view. In other words, the regret will increase because the DM is non-informative while the benchmark can predict accurately. Thus we can use the importance of an expert to bound the regret. The remaining question is, how to quantify the importance of an expert? We find that the benchmark function is a proper choice. We state our main result as Theorem E.6.

THEOREM E.6. *If $\Theta$ is $\alpha$-regular as defined in Definition E.11, for L2 loss function $\ell_2$, there exists a function $S(opt, k)$ depending on the*

*benchmark function opt and number of adversaries $k$ such that*

$$S(opt, k)^2 \geq R(\Theta, \Sigma) \geq \frac{\alpha}{4} S(opt, k)^2.$$

The theorem shows that we can only use the benchmark function to design a metric for the difficulty of adversarial information aggregation. Now we give the formula of $S(\cdot)$. First, we define the distance of reports.

DEFINITION E.7 (DISTANCE OF REPORTS). *For any vector $x$, we can define its histogram function $h_x(x) = \sum_i \mathbb{1}(x_i = x)$. For any pair of reports $x_1, x_2$ with the same size, their distance is defined by the total variation distance between $h_{x_1}(x)$ and $h_{x_2}(x)$.*

$$d(x_1, x_2) = TVD(h_{x_1}, h_{x_2}) = 1/2 \sum_x \left| h_{x_1}(x) - h_{x_1}(x) \right|.$$

*If $d(x_1, x_2) = 0$, we say that $x_1$ and $x_2$ are indistinguishable, which means they only differ in the order of reports. In fact, $d(x_1, x_2)$ is the minimal number of adversaries needed to make two reports indistinguishable. We also denote $d(x) = \sum_x h_x(x)$.*

LEMMA E.8. *Suppose the truthful experts will report $x_1 \in X^{n-k}$ or $x_2 \in X^{n-k}$. If and only if $k \geq d(x_1, x_2)$, there exists $x_A^1, x_A^2 \in X^k$ such that $d((x_1, x_A^1), (x_2, x_A^2)) = 0$. That is, $d(x_1, x_2)$ is the minimal number of adversaries needed to ensure the aggregator sees the same report vector $x$ in these two cases.*

PROOF OF LEMMA E.8. Consider the final report vector $x$ seen by the aggregator, we have $h_x(x) \geq \max(h_{x_1}(x), h_{x_2}(x))$ for any $x$. Thus to convert $x_2$ to $x$, we need at least

$$\sum_x |h_x(x) - h_{x_2}(x)| \geq \sum_x \max(h_{x_1}(x) - h_{x_2}(x), 0)$$
$$= 1/2 \sum_x \left( |h_{x_1}(x) - h_{x_2}(x)| + h_{x_1}(x) - h_{x_2}(x) \right)$$
$$= 1/2 \sum_x |h_{x_1}(x) - h_{x_2}(x)|$$
$$(\sum_x h_{x_1}(x) = \sum_x h_{x_2}(x) = n)$$

Similarly, to convert $x_1$ to $x$, we need $|h_{x_1}(x) - h_{x_2}(x)|$ adversaries, which complete our proof.

□

Now we define the sensitive parameter in the binary state case, where the benchmark function can be represented by a real number. Intuitively, it measures the greatest change $k$ experts can make regarding the benchmark function.

DEFINITION E.9 (SENSITIVE PARAMETER, BINARY). *When $|\Omega| = 2$, for any benchmark function opt, the sensitive parameter is defined by*

$$S(opt, k) = \max_{\theta, \theta' \in \Theta, d(x_T, x_T') \leq k} |opt_\theta(x_T) - opt_{\theta'}(x_T')|$$

We define the sensitive parameter in the binary state case. However, it is easy to generalize it to multi-state cases. We will discuss it later. Now we prove our main theorem.

PROOF OF THEOREM E.6. First, we construct a naive aggregator in the binary setting.

DEFINITION E.10 (NAIVE AGGREGATOR). *We use the average of the maximum and minimal prediction over all possible situations as the naive aggregator. Formally, we define* $u(x) = \max_\theta opt_\theta(x)$ *and* $l(x) = \min_\theta opt_\theta(x)$ *for any* $x \in X^{n-k}$. *The naive aggregator is*

$$f^{na}(x) = 1/2 \max_{d(x')=n-k, d(x,x')=k} u(x') + 1/2 \min_{d(x')=n-k, d(x,x')=k} l(x').$$

By the definition of $S(opt, k)$, for any $\theta \in \Theta, \sigma \in \Sigma$,

$$R(\theta, \sigma) \leq R(f^{na}, \theta, \sigma)$$
$$\leq \sup_{\theta \in \Theta, \sigma \in \Sigma} \mathbb{E}_{\theta, \sigma}[\ell(f^{na}(x), \omega)] - \mathbb{E}_\theta[\ell(opt_\theta(x_T), \omega)]$$
$$= \sup_{\theta \in \Theta, \sigma \in \Sigma} \mathbb{E}_{\theta, \sigma}[\ell(f^{na}(x), \omega) - \ell(opt_\theta(x_T), \omega)]$$
$$\leq \max_{\theta \in \Theta, \sigma \in \Sigma} \ell(|f^{na}(x) - opt_\theta(x_T)|)$$
$$\leq \ell(S(opt, k)/2)$$

However, The proof fails in the multi-state case since $f^{na}(x)$ is not a distribution, thus illegal in this setting. Instead, we need another aggregator.

Suppose $u(x)_\omega = \max_\theta opt_\theta(x)_\omega$. Then it is obvious that $\sum_\omega u(x)_\omega \geq \sum_\omega opt_\theta(x)_\omega = 1$. Similarly we have $l(x)_\omega = \min_\theta opt_\theta(x)_\omega$ and $\sum_\omega l(x)_\omega \leq \sum_\omega opt_\theta(x)_\omega = 1$.

Thus for every $x$ there exists a vector $f'(x)$ such that for any $\omega \in \Omega, l(x)_\omega \leq f'(x)_\omega \leq u(x)_\omega$ and $\sum_\omega f(x)_\omega = 1$. We can pick $f'(x)$ as the aggregator and for any $\theta \in \Theta, \sigma \in \Sigma$,

$$R(\theta, \sigma) \leq R(f', \theta, \sigma)$$
$$\leq \sup_{\theta \in \Theta, \sigma \in \Sigma} \mathbb{E}_{\theta, \sigma}[\ell(f'(x), \omega)] - \mathbb{E}_\theta[\ell(opt_\theta(x_T), \omega)]$$
$$= \sup_{\theta \in \Theta, \sigma \in \Sigma} \mathbb{E}_{\theta, \sigma}[\ell(f'(x), \omega) - \ell(opt_\theta(x_T), \omega)]$$
$$\leq \max_{\theta \in \Theta, \sigma \in \Sigma} \ell(|f'(x) - opt_\theta(x_T)|)$$
$$\leq \ell(S(opt, k))$$

Now we consider the lower bound of $R(\Theta, \Sigma)$ (Lemma E.12). A basic idea is that we can construct a coupling of two information structures whose reports have a distance smaller than $k$. Then the adversaries can make the coupling indistinguishable. However, if the probability of the worst case is too small, then their contribution to the regret is also negligible. Thus we need to constrain the information structures (Definition E.11).

DEFINITION E.11 ($\alpha$-REGULAR INFORMATION STRUCTURE). *An information structure* $\theta$ *is* $\alpha$-*regular if every possible report vector* $x$ *will appear with probability at least* $\alpha$:

$$\min_{x: \Pr_\theta[x] > 0} \Pr_\theta[x] > \alpha.$$

LEMMA E.12. *If* $\theta$ *is* $\alpha$-*regular, for any benchmark function opt and L2 loss* $\ell_2$, $R(\Theta, \Sigma) \geq \frac{\alpha}{4} S(opt, k)^2$.

PROOF. Fix $\alpha$, let the maximum of $S(opt, k)$ is obtained by $\theta, \theta', x_T, x'_T$. We construct an information structure as follows. Suppose the joint distribution $\theta''$ is the mixture of $\theta$ and $\theta'$ with equal probability. By the definition of $d(x_T, x'_T)$, there exists $\sigma'$ such that $h_{x_T, \sigma'}(x_T) = h_{x'_T, \sigma'}(x'_T)$. Then we have

$$R(\theta, \sigma) \geq R(\theta'', \sigma')$$
$$\geq \min_f \mathbb{E}_{\theta'', \sigma'}[\ell(f(x), \omega)] - \mathbb{E}_{\theta''}[\ell(opt''_\theta(x''_T), \omega)]$$
$$= \min_f \mathbb{E}_{\theta'', g'}[(f(x) - opt_{\theta''}(x''_T))^2]$$
$$= 1/2 \min_f \Pr_\theta[x_T](f(x) - opt_\theta(x_T))^2 + \Pr_{\theta'}[x'_T](f(x) - opt_{\theta'}(x'_T))^2$$
$$\geq \frac{\Pr_\theta[x_T] \Pr_{\theta'}[x'_T]}{2(\Pr_\theta[x_T] + \Pr_{\theta'}[x'_T])}(opt_\theta(x_T) - opt_{\theta'}(x'_T))^2$$
$$\geq \frac{\alpha}{4} S(opt, k)^2$$

□

Combining the lower bound and upper bound, we infer Theorem E.6.

□

Received 20 February 2007; revised 12 March 2009; accepted 5 June 2009

