# OpenReview forum: "Robust Aggregation with Adversarial Experts"
_ACM.org/TheWebConf/2025/Conference — WWW 2025 Poster_

### Official Review · Reviewer_ND1x · 2024-11-20

**Novelty:** 4
**Technical Quality:** 4

**Review:**

The paper addresses a robust aggregation problem involving both truthful and adversarial experts. It proposes optimal aggregators for various binary states and provides a theoretical analysis for scenarios where an optimal aggregator cannot be determined. Additionally, the article includes experimental verification using CIFAR-10 in the context of ensemble learning. While the problem explored in the article is intriguing and the main content is relatively clear, there are some deficiencies in both the content organization and the comprehensiveness of the experiments.

Pros：
1. The paper delves into fundamental theoretical issues, providing a comprehensive analysis that spans a broad range of fields.
2. The contributions of the paper are clearly enumerated, offering a structured overview of the novel insights and advancements presented.
3. The theoretical work presented in the paper is supported by rigorous proofs, ensuring the validity and reliability of the conclusions drawn.
4. Several numerical experiments were conducted to complement the theoretical analysis. These experiments serve to validate the theoretical models and provide empirical evidence that supports the paper's claims.

Cons:
1. The main body of the article is seven pages long, which is insufficient for a self-contained paper.
2. The baselines used for experimental comparison are relatively simplistic, consisting only of voting and averaging. As noted in the article, there are numerous more advanced algorithms applicable to the ensemble learning scenario. However, the adaptation of these methods to this experimental context is not discussed and verified.
3. The extension to the multi-state case is a crucial discussion, as it relates to the significance of the theory proposed in the paper and its implications for future research. It is recommended that this discussion be included in the main body of the article.
4. Minor issues include the organization of the article's content, such as overly lengthy formulas (e.g., Theorem 4.2, and Equation (2)).

**Questions:**

1. The main body of the article is seven pages long, which is insufficient for a self-contained paper.
2. The baselines used for experimental comparison are relatively simplistic, consisting only of voting and averaging. As noted in the article, there are numerous more advanced algorithms applicable to the ensemble learning scenario. However, the adaptation of these methods to this experimental context is not discussed and verified.
3. The extension to the multi-state case is a crucial discussion, as it relates to the significance of the theory proposed in the paper and its implications for future research. It is recommended that this discussion be included in the main body of the article.
4. Minor issues include the organization of the article's content, such as overly lengthy formulas (e.g., Theorem 4.2, and Equation (2)).

**Reviewer Confidence:**

3: The reviewer is confident but not certain that the evaluation is correct

**Scope:**

3: The work is somewhat relevant to the Web and to the track, and is of narrow interest to a sub-community

---

### Official Review · Reviewer_mumW · 2024-12-01

**Novelty:** 6
**Technical Quality:** 6

**Review:**

Originality:

The authors extend the robust information aggregation framework by considering adversarial experts, as opposed to the previous work that studied only truthful but possibly unreliable experts. This adversarial setting requires developing new theoretical tools and analytic techniques in characterizing the optimal aggregators and their regrets.

Significance:

The results in this paper have significant implications for designing robust aggregation mechanisms in practical applications when adversarial manipulations are a concern. The fact that the performance of the truncated mean is theoretically optimal under L1 loss greatly justifies its extensive practical use. Optimal closed-form aggregators are obtained for both adversarial and non-adversarial cases under L2 loss; this gives much weight to the theoretical value of this problem.

Clarity:

The paper is generally well-written, flowing well to present its ideas in logical order. The examples, figures, and tables used to explain various concepts effectively make complex ideas clearer and more understandable from a theoretical point of view. Table 1 summarizes the results for quick reference of the key contributions of the paper.

Quality:

The paper is technically sound and has well-presented proofs and derivations. The numerical experiments in Section 5 give an empirical validation of the theoretical results of a realistic ensemble learning task. The discussion of limitations and possible extensions in Section 6 and Appendix E shows that the authors are well aware of the scope of the work and possible avenues for further research.


Pros:

Novel and practically relevant problem setting: Introducing adversarial experts significantly extends the scope of robust information aggregation research.
Strong theoretical results: The paper provides provable guarantees of the optimality of various aggregation mechanisms, including the commonly used truncated mean.
Empirical validation: Numerical experiments show that the proposed aggregators perform well in a practical setting relevant to the application domain.

Cons:

Assumptions and limitations: The paper is based on certain assumptions, such as the symmetry of experts and knowledge of the adversary ratio.

Overall, this work represents a valuable contribution to robust information aggregation. Its novel problem setting, rigorous theoretical analysis, and empirical validation make it a significant advancement in understanding and addressing the challenges of aggregating information from potentially malicious sources.

**Questions:**

The paper states that L2 loss seems to enforce much more conservative strategies than L1 loss. The numerical experiments demonstrate that L2 loss performs poorly compared to L1. Could the authors provide more details on why L2 loss enforces conservative behavior?

**Reviewer Confidence:**

2: The reviewer is willing to defend the evaluation, but it is likely that the reviewer did not understand parts of the paper

**Scope:**

3: The work is somewhat relevant to the Web and to the track, and is of narrow interest to a sub-community

---

### Official Review · Reviewer_Vb5W · 2024-12-03

**Novelty:** 4
**Technical Quality:** 4

**Review:**

**Summary**

1. The authors consider the problem of robust aggregation in the presence of honest and adversarial experts. Honest experts report their private signals truthfully, while adversarial experts can report arbitrarily. The goal is to find the best aggregator that outputs predictions that minimize regret under the worst information structure and adversarial strategy.

2. Under L1 loss, the authors show that the truncated mean aggregator is optimal. For L2 loss, the best aggregator is a piecewise linear function.

3. For hard aggregators that output decisions, the authors show that a random version of the truncated mean is optimal for both L1 and L2.

4. The authors numerically evaluate their aggregator on an ensemble learning task.

**Strengths**

1. The authors provide detailed proofs for their theoretical results and conduct extensive numerical experiments to validate their findings.

2. The significance of this work lies in its practical applications and theoretical advancements. The robust aggregation problem is relevant to various domains, including crowdsourcing, ensemble learning, and election systems. The authors' findings have the potential to improve the reliability of information aggregation in these domains, particularly in the presence of adversarial attacks.

**Weaknesses**

1. The authors clearly define their problem statement, theoretical results, and numerical experiments. However, some sections, particularly the proofs in the appendices, could be challenging for readers without a strong background in theoretical computer science or information design.

2. The paper focuses on binary aggregation and could be extended to consider more complex decision-making scenarios.

**Questions:**

1. Can your framework be extended to consider non-binary decisions? If so, what additional complexities or challenges would arise?

2. Have you considered extending your framework to other loss functions beyond L1 and L2? If so, what are the key challenges and potential benefits?

3. Can you provide more insights on aggregators design in the paper?

**Reviewer Confidence:**

2: The reviewer is willing to defend the evaluation, but it is likely that the reviewer did not understand parts of the paper

**Scope:**

3: The work is somewhat relevant to the Web and to the track, and is of narrow interest to a sub-community

---

### Official Review · Reviewer_CAxy · 2024-12-03

**Novelty:** 4
**Technical Quality:** 4

**Review:**

This paper considers the prediction aggregation problem with the presence of adversarial experts. The paper analyze the cases under L1 and L2 loss and derive a closed-form description of aggregation rule, 𝑘-truncated mean. The result relies on the knowledge of the number of adversaries among the experts, k. The paper also presents an experiment of ensemble learning task that aggregates the prediction of multiple image classifiers.

I have little experience with the prediction aggregation problem, but the paper very well educates me the basics. The results in this paper seems to be solid, though I did not go into the details of its proof. The focus on binary outcomes seem to be natural in many cases, but it would be interesting to understand how the current results could generalize to the case of multiple outcomes.

Typo: capture in Figure 3

**Questions:**

Please see my questions in the review above.

**Reviewer Confidence:**

3: The reviewer is confident but not certain that the evaluation is correct

**Scope:**

3: The work is somewhat relevant to the Web and to the track, and is of narrow interest to a sub-community

---

### Official Review · Reviewer_vqAV · 2024-12-04

**Novelty:** 5
**Technical Quality:** 6

**Review:**

Summary : This paper studies the Robust Aggregation problem in the presence of both adversarial and truthful experts where all the experts share symmetric marginal distributions i.e., same prior and posteriors.  Especially, their focus is on binary states and reports. Their main contributions include (i) optimal theoretical for the regret minimization problem under L1 and L2 losses under the worse case information structure and adversarial reporting , (ii) specifically, for L1 loss, they show that the k-truncated mean is the optimal where k is the number of adversaries. (iii) for L2 loss, they show that the optimal aggregator is a piece wise linear function for which they give a closed form solution. All the bounds are conditional on the ratio of adversaries vs truthful experts which depends on the prior and posterior distributions.


Pros :
1. The paper is mostly well written and the theoretical results looks interesting. Especially, it's surprising to see that for the case of L2 loss, there is indeed a closed form solution in the presence of adversaries.
2. Their aggregators perform better than the traditional heuristics like majority vote and averaging in the case of L2 loss.

Cons:
1. I think having examples in the Sections 1,3 can greatly improve readability. It was hard to follow the paper without clear examples in mind especially since there is a lot of technical jargon.
2. There is not enough discussion about the performance of the majority vote aggregator consistently being close to the optimal aggregators for L1 loss. Can you give some reasons why especially even when k is relatively large.

**Questions:**

Please see weaknesses.

**Reviewer Confidence:**

2: The reviewer is willing to defend the evaluation, but it is likely that the reviewer did not understand parts of the paper

**Scope:**

4: The work is relevant to the Web and to the track, and is of broad interest to the community